energy/engineering geology/geophysics

failure depth of floor rocks, orthogonal numerical tests, impact factor, prediction model, *in situ* monitoring, sensitivity analysis

**Author for correspondence:**
Xiaoqiang Zhang
e-mail: tyzxq2009@163.com

# A multifactor coupling prediction model for the failure depth of floor rocks in fully mechanized caving mining: a numerical and *in situ* study

Yulong Jiang[1,3], Tingting Cai[2] and Xiaoqiang Zhang[3]

[1]Key Laboratory of In-situ Property Improving Mining of Ministry of Education,
[2]College of Safety and Emergency Management Engineering, and [3]College of Mining Engineering, Taiyuan University of Technology, Taiyuan, Shanxi 030024, People's Republic of China

 YJ, 0000-0002-3747-503X

To study the mining-induced failure depth of floor rocks in a fully mechanized mining caving field affected by different coal seam pitches, mining face lengths, burial depths and aquifer water pressures, multifactor-coupled orthogonal numerical tests on the failure depth of floor rocks were conducted. The numerical results show that the failure depth of floor rocks increases with increasing mining face length, coal seam pitch and burial depth. According to the relationship between failure depth and these impact factors, a multifactor-coupled prediction model for the failure depth of floor rocks was established. In addition, the *in situ* measurement of the failure depth of floor rocks in the Yitang Coal Mine in Huoxi coal field in Shanxi Province, China, was performed, and the *in situ* failure depths of floor rocks in the 100 502 (80 m) and 100 502 (180 m) mining faces were approximately 12.50–14.65 m and 17.50–19.20 m, in good agreement with the results of the multifactor prediction model. Furthermore, the sensitivity of each impact factor in the prediction model of the floor failure depth was further analysed by *F*-test and range analysis, and the impact order of studied factors on the floor failure depth is coal seam pitch > mining face length > burial depth > aquifer water pressure.

## 1. Introduction

The Carboniferous–Permian coalfield in North China is one of the most productive coal fields in China, where mining has transferred

to deep coal seams due to increasing energy demand despite the instability of the Ordovician limestone aquifer in the deep rocks. When mining under the aquifer water pressure, the preventions against the connection of the Ordovician limestone aquifer and the damaged floor rocks are the basic insurances to avoid water inrush disasters [1]. Therefore, it is of great significance to study the failure of floor rocks to ensure safe and efficient mining under the water pressure of aquifers.

Currently, many studies on the failure of floor rocks have been reported since a critical energy-release point was introduced by Santos & Bieniawski to analyse the bearing capacity and stability principles of the floor rocks in long-term conditions [2,3]. Kumar and Das carried out plate loading tests on simulated floor strata for varying geotechnical conditions of weak floor strata to evaluate the bearing strength characteristics of floor strata during an excavation process while under heavy loads from an overlying rock mass [4]. The failure depth in a specific longwall gangue backfilling mine was measured using the mine electricity profiling method by Yang *et al.* [5]. Yin *et al.* performed field monitoring of the stress state of the floor rocks during the mining process and revealed the relationship between the vertical/ horizontal stress increments of the floor rocks with depth [6,7]. Based on the isotropic assumption of rocks, Zhang determined the failure depth of floor rocks by comparing the flow rate of water in a borehole before and after coal mining [8]. In combination with the methods of field measurement, theoretical analysis and numerical simulation, Zhu *et al.* [9] and Wang *et al.* [10] developed analytical solutions of the support stress distribution of floor rocks to quantify the failure depths of floor rocks. Jiang *et al.* performed an *in situ* measurement by hollow inclusion strain sensors to investigate the mining-induced damage and failure characteristics of coal seam floor rocks at different depths in working faces with different lengths [11]. Generally, among the above studies, many impact factors, such as *in situ* stress, burial depth and isotropic rocks, were taken into account [12–15], while a few studies that include aquifer water have been reported. The failure depth of the floor rocks affected by the mining process is mainly composed of two parts: the direct failure zone formed by the mining stress field and the floor rising zone affected by the aquifer water. The sum of the two is the final failure depth of the floor rocks [16]. However, the effect of aquifer water on the direct failure zone is always ignored. The high pressure of the aquifer water will cause structural changes in the floor rocks and can eventually lead to the changes in failure depth of floor rocks [17]. Therefore, the influence of the aquifer water on the failure depth of the floor rocks cannot be ignored.

The *in situ* measurement of the stress or strain of floor rocks under active mining is the most direct and effective method for determining the deformation and failure of floor rocks [18–20]. However, limited by the specific and non-repeatable geophysical conditions in *in situ* monitoring, only certain impact factors could be considered in field measurements. The failure of floor rocks is always the result of several impact factors, and numerical testing is an effective method to take multifactor coupling into consideration. It is of great significance to study the failure depth of floor rocks affected by multiple impact factors and give the impact order of the factors.

In this paper, multifactor-coupled orthogonal numerical tests on the failure depth of floor rocks were performed, the impact of these factors on the failure depth was analysed, and a mathematical multifactor prediction model for the failure depth of floor rocks was established. Furthermore, this prediction model was validated by the *in situ* monitoring results of the floor rocks in the Yitang coal mine, a sensitivity analysis of each impact factor on the failure depth was performed, and the impact order of the factors on floor rocks failure depth was determined. The results provide important guidance for coal mining under water pressure and offer a key theoretical reference for the failure depth control of floor rocks under similar geophysical conditions.

# 2. Multifactor-coupled numerical tests of floor rock failure depth

## 2.1. Numerical tests

Given the limitations of the specific and non-repeatable geophysical conditions, multifactor-coupled numerical tests of floor rock failure depth were conducted. The impact factors, burial depth, mining face length, coal seam pitch and aquifer water pressure were taken into consideration. In the numerical tests, a three-dimensional fluid–solid coupling numerical simulation model was established (figure 1).

In this model, the rock strata are divided into grids by 8-node parametric elements, and the Coulomb–Molar criterion is considered. The mechanical parameters of the strata floor rocks are measured from the core samples from *in situ* drilling and are shown in table 1. In the numerical simulation tests, the impact of burial depth (350 m, 400 m, 450 m and 500 m), mining face length (80 m, 120 m, 150 m and 180 m),

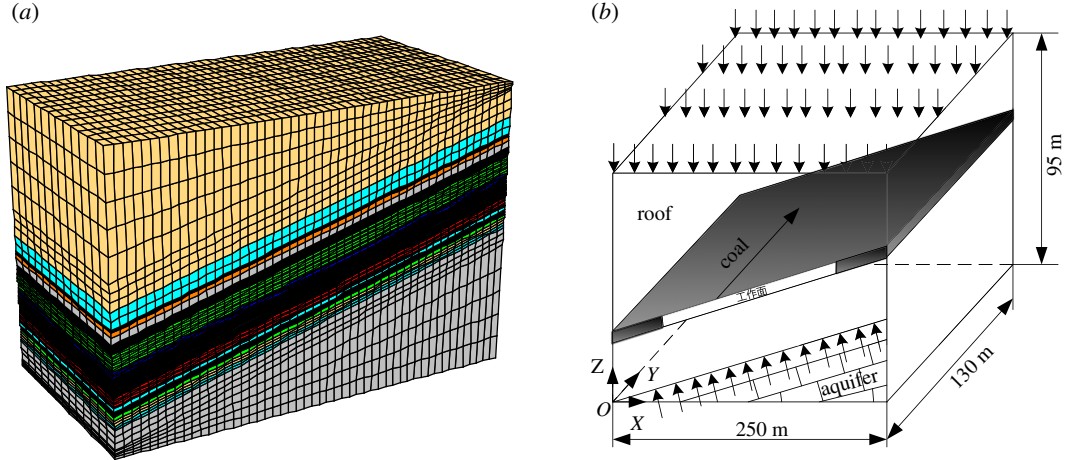

**Figure 1.** Three-dimensional fluid–solid coupled numerical simulation model.

**Table 1.** Lithology and mechanical parameters of the floor strata.

| borehole length (m) | rock | tensile strength (MPa) | compressive strength (MPa) | cohesion (MPa) | Poisson ratio | elastic modulus (GPa) |
|---|---|---|---|---|---|---|
| — | coal | 0.75 | 7.78 | 1.13 | 0.25 | 2.50 |
| 2.30–3.57 | mudstone | 3.57 | 25.58 | 1.98 | 0.29 | 3.72 |
| 3.57–3.70 | fine-grained sandstone | 6.45 | 62.47 | 2.02 | 0.27 | 11.07 |
| 3.70–4.70 | sandy mudstone | 3.49 | 32.63 | 2.08 | 0.28 | 3.48 |
| 4.70–7.70 | mudstone | 3.57 | 25.58 | 1.98 | 0.29 | 3.72 |
| 7.70–20.60 | sandy mudstone | 3.49 | 32.63 | 2.08 | 0.28 | 3.48 |
| 20.60–22.50 | sandstone | 6.45 | 42.47 | 2.17 | 0.21 | 5.72 |

coal seam pitch (0°, 5°, 10° and 15°) and aquifer water pressure (0 MPa, 1.5 MPa, 3 MPa and 4.5 MPa) on the failure depth of floor rocks are taken into account. Considering that the maximum mining face is 180 m and the maximum coal seam pitch is 15°, the size of the numerical simulation model is length × width × height = 250 m ($x$) × 130 m ($y$) × 95 m ($z$).The bottom of the model is constrained by vertical displacement, the front, back, left and right sides of the model are constrained by horizontal displacement, and the upper surface is in a free state. The overlying rock strata (except the coal seam roof) are uniformly loaded to the upper surface. The aquifer water pressure is loaded to the coal seam at a rate of −0.01 MPa m$^{-1}$ downward along the coal seam. In this numerical simulation, the orthogonal numerical simulation schemes are designed as shown in table 2.

## 2.2. Numerical results of the failure depth of floor rocks

Figure 2 shows the distribution map of the damaged floor rocks along the coal seam pitch with a mining height of 7.7 m from these numerical simulation tests. As shown in figure 2, shear failure is the main failure mode of the floor rocks, and the failure of the floor rocks increases gradually with the increasing mining face length, coal seam pitch and burial depth, while the effect of the aquifer water pressure on failure is not obvious.

The failure depths of the floor rocks in these numerical tests are collected and shown in table 3. Because four-factor orthogonal numerical tests are performed, the failure depths in the schemes with the same factor should be calculated by averaging when analysing the failure depth of the floor rocks for each impact factor. For example, the mining face lengths of schemes #1, 2, 3 and 4 are all 80 m, and their failure depths are 7.65 m, 9.94 m, 10.97 m and 18.77 m, respectively. The average failure depth is 11.83 m, corresponding to the failure depth at the mining face length of 80 m without considering other impact factors. Similarly, the mining face lengths of schemes #5, 6, 7 and 8 are all

**Table 2.** The orthogonal numerical simulation schemes.

| test number | test schemes | | | |
| --- | --- | --- | --- | --- |
| | mining face length (m) | coal bed pitch (°) | burial depth (m) | aquifer water pressure (MPa) |
| #1 | 80 | 0 | 350 | 0 |
| #2 | 80 | 5 | 400 | 1.5 |
| #3 | 80 | 10 | 450 | 3 |
| #4 | 80 | 15 | 500 | 4.5 |
| #5 | 120 | 0 | 400 | 3 |
| #6 | 120 | 5 | 350 | 4.5 |
| #7 | 120 | 10 | 500 | 0 |
| #8 | 120 | 15 | 450 | 1.5 |
| #9 | 150 | 0 | 450 | 4.5 |
| #10 | 150 | 5 | 500 | 3 |
| #11 | 150 | 10 | 350 | 1.5 |
| #12 | 150 | 15 | 400 | 0 |
| #13 | 180 | 0 | 500 | 1.5 |
| #14 | 180 | 5 | 450 | 0 |
| #15 | 180 | 10 | 400 | 4.5 |
| #16 | 180 | 15 | 350 | 3 |

120 m, and their failure depths are 12.17 m, 10.07 m, 18.64 m and 18.79 m, respectively. The average failure depth is 14.92 m, corresponding to the failure depth at the mining face length of 120 m. Consequently, the failure depths of floor rocks at each impact factor are calculated and shown in table 4.

Combining the above numerical simulation results of the failure depths, the impacts of the four factors on the failure depths of the floor rocks are analysed as follows.

(a) The average failure depths with mining face lengths of 80 m, 120 m, 150 m and 180 m are 11.83 m, 14.92 m, 17.18 m and 17.17 m, respectively. The failure depths of the floor rocks increase with the mining face length, but when the mining face length exceeds 150 m, the increase in the failure depth is not obvious.

(b) The average failure depths with coal seam pitches of 0°, 5°, 10° and 15° are 11.03 m, 14.52 m, 16.82 m and 18.73 m, respectively. The failure depths of floor rocks increase with increasing coal seam pitch. In addition, when the coal seam pitch is greater than 0°, the failure scope of the two sides of the mining face is smaller than that of the centre; on the contrary, the failure scope of the two sides of the mining face is larger than that of the centre when the coal seam pitch is 0°.

(c) The average failure depths with burial depths of 350 m, 400 m, 450 m and 500 m are 13.87 m, 14.86 m, 15.22 m and 17.17 m, respectively. The failure depths of the floor rocks increase with increasing coal seam burial depth.

(d) The average failure depths with an aquifer water pressure of 0 MPa, 1.5 MPa, 3 MPa and 4.5 MPa are 15.94 m, 14.94 m, 15.28 m and 14.95 m, respectively, suggesting that the increasing aquifer water pressure has little direct impact on the failure depth of the floor rocks.

# 3. Multifactor prediction model for the failure depth of floor rocks

According to the above numerical simulation results of the failure depths of the floor rocks, the relationship between the failure depths and the impact factors (mining face length, coal bed pitch, burial depth and aquifer water pressure) are quantitatively shown in figure 3.

As shown in figure 3, the mining face length, coal seam pitch and aquifer water pressure are in good logarithmic relation with the failure depths of the floor rocks, and the corresponding correlation coefficients are 0.9603, 0.9907 and 0.9881, respectively. The burial depth of the coal seam is in a good exponential relation with the failure depth of the floor rocks, with a correlation coefficient of 0.9297.

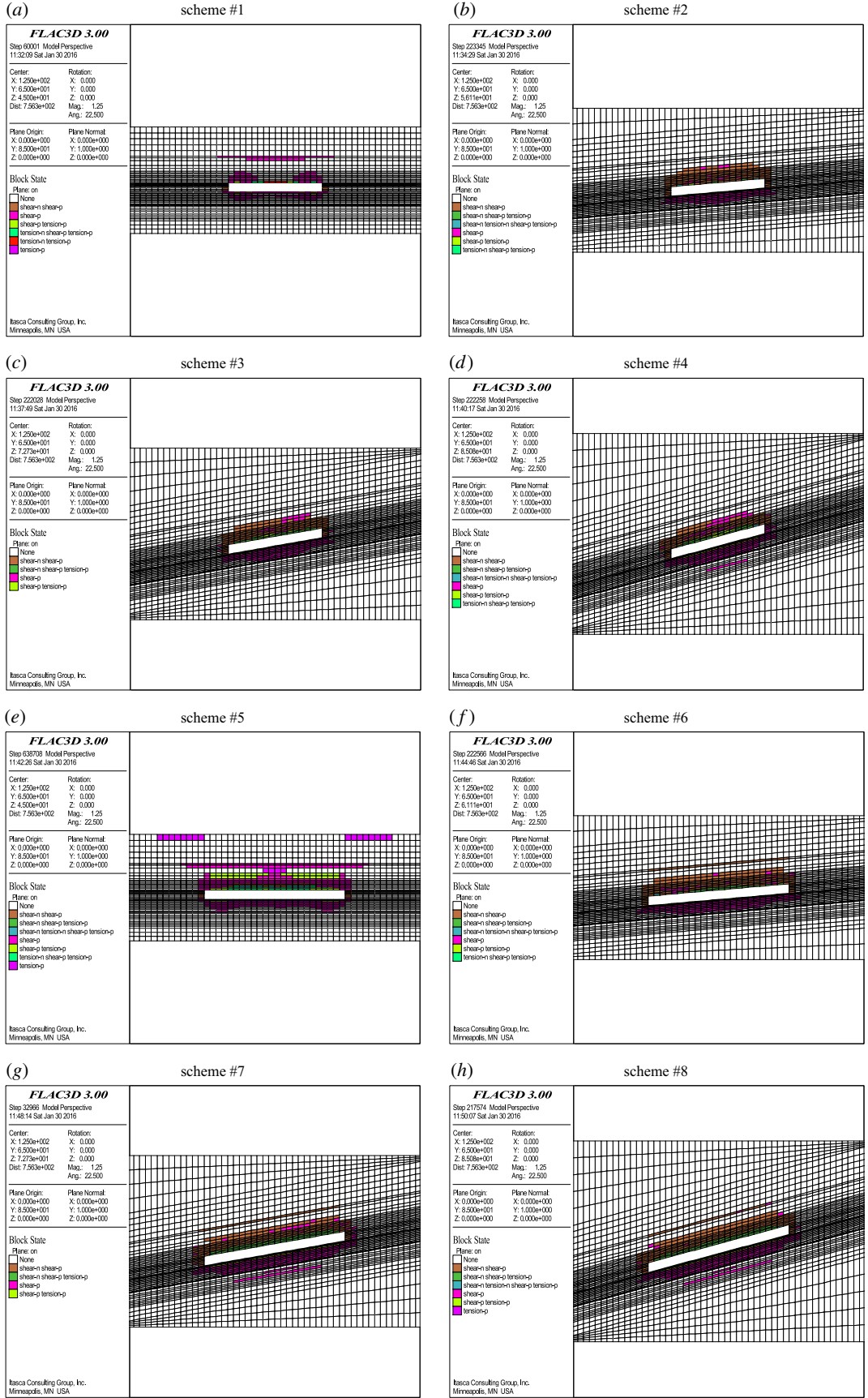

**Figure 2.** Distribution map of the damaged floor rocks along the coal seam pitch for each numerical simulation scheme.

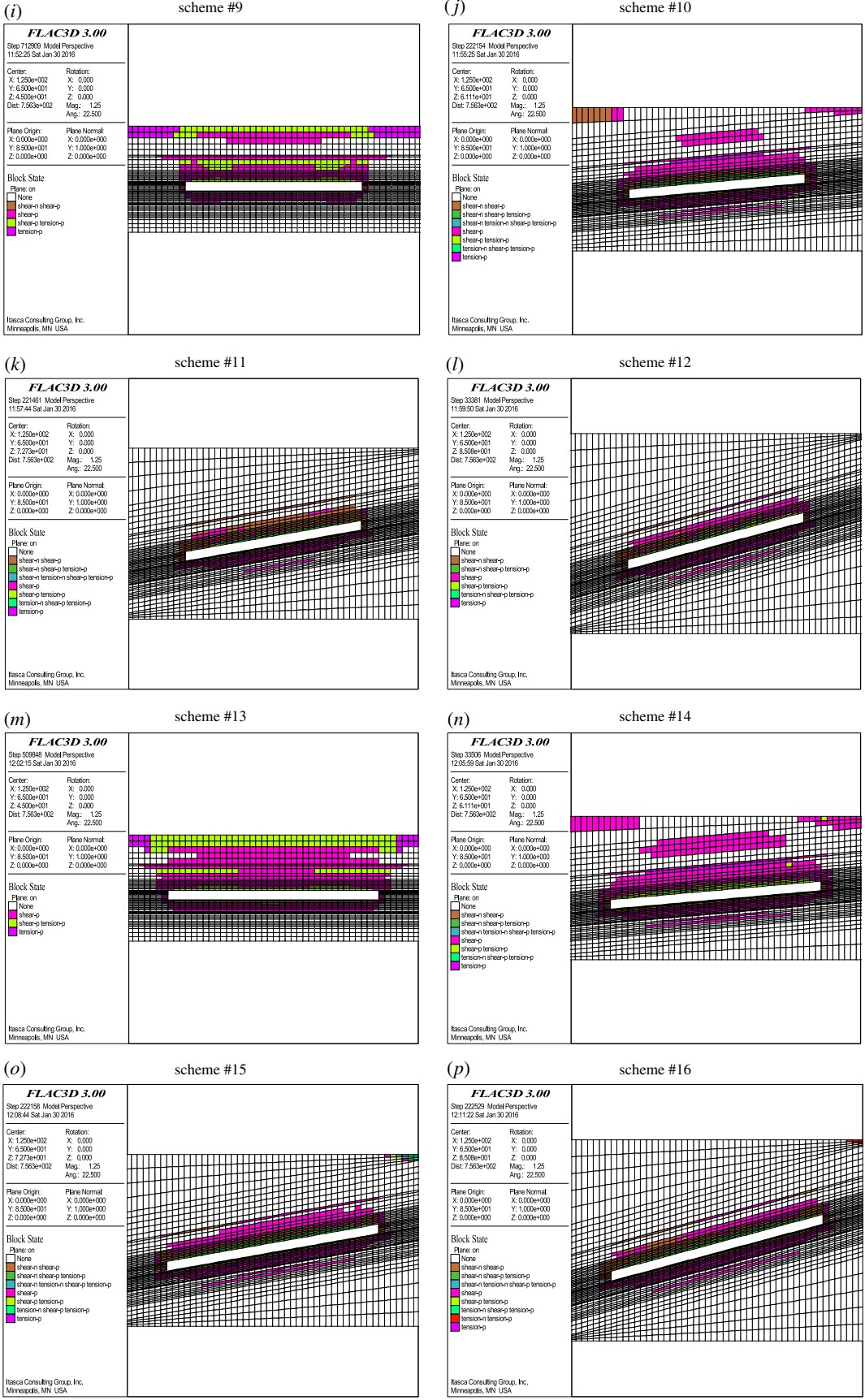

**Figure 2.** (*Continued.*)

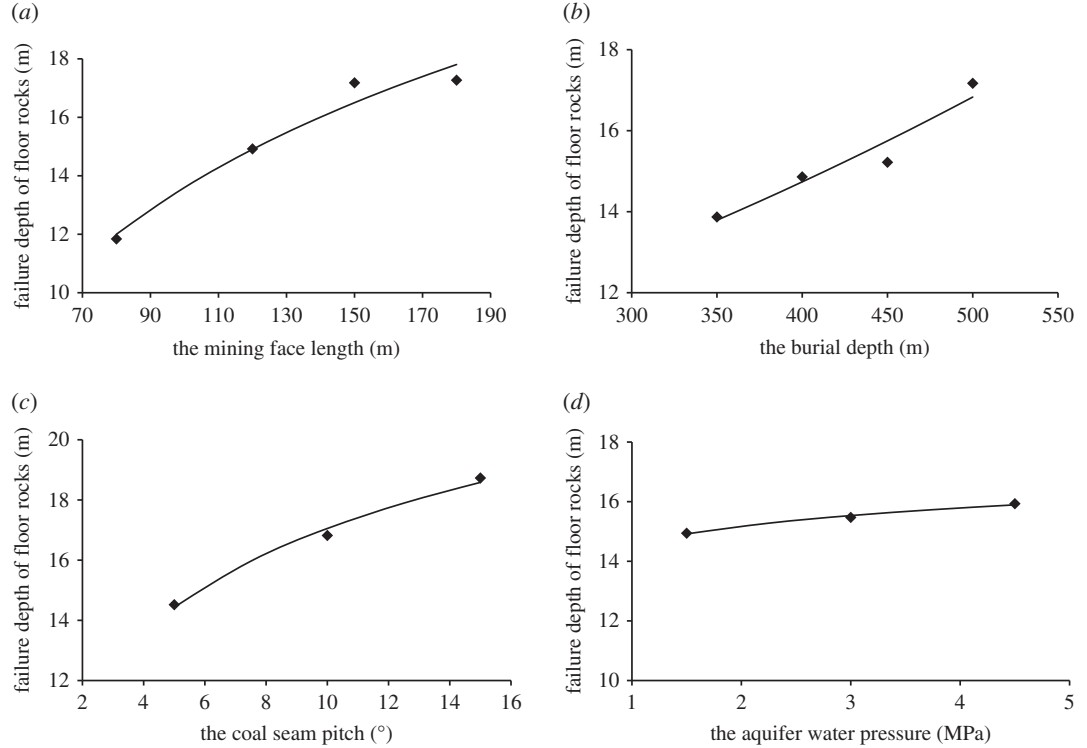

**Figure 3.** Relationship between failure depth and the impact factors.

**Table 3.** Failure depths of floor rocks in each numerical test.

| schemes | failure depths (m) | schemes | failure depths (m) |
| --- | --- | --- | --- |
| #1 | 7.65 | #9 | 12.17 |
| #2 | 9.94 | #10 | 19.12 |
| #3 | 10.97 | #11 | 18.90 |
| #4 | 18.77 | #12 | 18.53 |
| #5 | 12.17 | #13 | 12.13 |
| #6 | 10.07 | #14 | 18.94 |
| #7 | 18.64 | #15 | 18.78 |
| #8 | 18.79 | #16 | 18.84 |

**Table 4.** Average failure depth of floor rocks at each impact factor.

| impact factor | failure depths (m) | impact factor | failure depths (m) |
| --- | --- | --- | --- |
| mining face length of 80 m | 11.83 | burial depth of 350 m | 13.87 |
| mining face length of 120 m | 14.92 | burial depth of 400 m | 14.86 |
| mining face length of 150 m | 17.18 | burial depth of 450 m | 15.22 |
| mining face length of 180 m | 17.17 | burial depth of 500 m | 17.17 |
| coal bed pitch of 0° | 11.03 | aquifer water pressure of 0 MPa | 15.94 |
| coal bed pitch of 5° | 14.52 | aquifer water pressure of 1.5 MPa | 14.94 |
| coal bed pitch of 10° | 16.82 | aquifer water pressure of 3.0 MPa | 15.28 |
| coal bed pitch of 15° | 18.73 | aquifer water pressure of 4.5 MPa | 14.95 |

**Table 5.** Multiple linear regression analysis results.

| source | sum of squares | degree of freedom | mean square | F-value | significance F |
|---|---|---|---|---|---|
| regression | 11 | 71.83756 | 50.60076 | 7.748152 | 0.003194 |
| residual | 4 | 202.403 | 6.530687 | | |
| total | 15 | 274.2406 | | | |

Therefore, a multifactor linear regression expression is used to quantitatively analyse the failure depth, and the expression is as follows:

$$h = A \ln L + B \ln \alpha + C \ln P + De^{mH} + E, \tag{3.1}$$

where $h$ is the failure depth of the floor rocks, $L$ is the mining face length, $\alpha$ is the coal seam pitch, $P$ is the aquifer water pressure, $H$ is the burial depth, and $A$, $B$, $C$, $D$, $E$ and $m$ are coefficients.

The relationships between the failure depths of the floor rocks and the impact factors are substituted into equation (3.1), and the corresponding linear regression is taken, so equation (3.1) can be rewritten as

$$h = -45.8526 + 7.074 \ln L + 5.504 \ln \alpha + 9.1214 e^{0.0013H} - 0.4079 \ln P. \tag{3.2}$$

The multiple linear regression analysis results are shown in table 5. The $F$-value of equation (3.2) is 7.748152, and the significance of $F$ is 0.003194, far smaller than 0.05, suggesting that there is a good regression relationship between the failure depth and the impact factors (mining face length, coal seam pitch, aquifer water pressure and burial depth) and that the newly established multifactor prediction model for the failure depth of floor rocks is of statistical significance.

# 4. Model validation

To check the accuracy of the multifactor prediction model for the failure depth of floor rocks, the *in situ* permissions were obtained from the Yitang coal mine, Shanxi, China and the *in situ* measurements of the failure depth of the floor rocks in the Yitang coal mine were conducted, and the *in situ* results were used for model validation.

## 4.1. *In situ* monitoring

In field monitoring, the strain sensors are buried in holes at different depths, and the strain evolutions are recorded during the mining process. Once the surrounding rocks at the buried positions are disturbed violently, strong plastic displacement and deformation of the drilling hole inevitably occur and damage the plastic coating of the probe or even cause malfunctions of the probe; thus, the measured strains are highly discrete or fluctuate randomly to reflect the damaged floor rocks [11].

The area for *in situ* field measurement is located in the no. 100502 fully mechanized caving mining face of the Yitang Coal Mine in Huoxi coal field in Shanxi Province of China. The #9 coal seam in the minefield is quite complex, with a high coal seam pitch and many faults; in particular, in the northeast of the minefield, the coal seam burial depth is deep and the aquifer water pressure is high. The coal seam has a depth of 500 m and an average thickness of 7.7 m. The elevation of the #9 coal seam floor is 120–620 m. The elevation of the Ordovician limestone aquifer in the minefield is 531.5–530 m. When the elevation of the coal seam floor is 200–255 m, the water inrush coefficient is greater than 0.1 MPa m$^{-1}$, and the maximum water inrush coefficient is 0.151 MPa m$^{-1}$. The risk of water inrush in this minefield is extremely great. The layout of the no. 100502 mining faces is shown in figure 4a. The mining roadways are arranged along the floor. The roof of the coal seam is limestone with a thickness of 11 m. The coal seam floor is dominated by mudstone, carbonaceous mudstone and sandy mudstone. In this field monitoring, the *in situ* strain induction method was used to determine the failure depth of the floor rocks according to the deformation of the floor rocks at different burial depths. Four monitoring boreholes were drilled in the mining face, and 17 hollow body monitoring probe sensors were arranged in the four monitoring holes, as shown in figure 4b. Each monitoring hole was arranged at an angle of 30° to the coal seam floor, and these probe sensors were placed in the boreholes at an interval along their depths to more accurately monitor the stress state of the rocks at different depths.

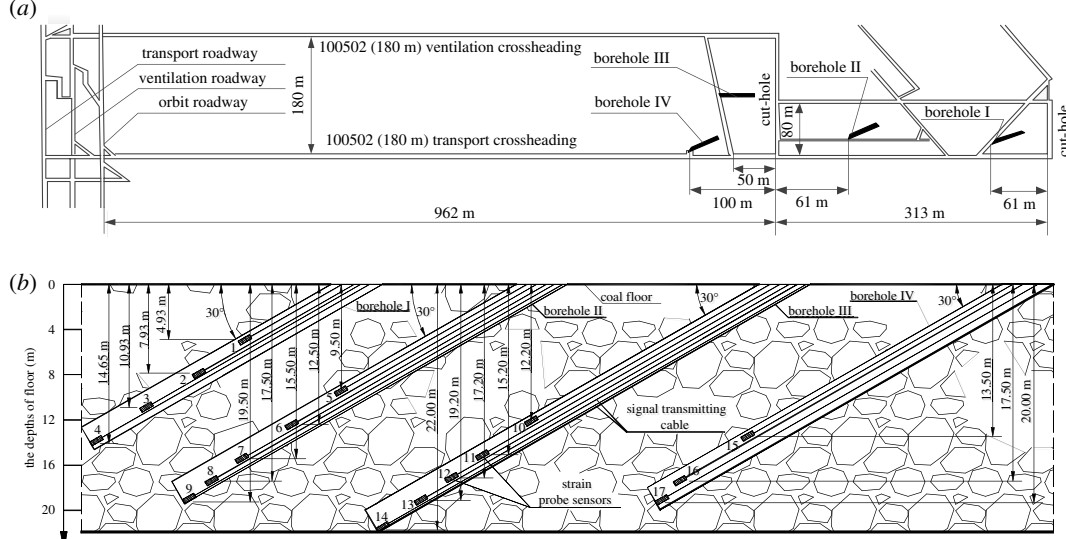

**Figure 4.** The layout of no. 100 502 mining face.

The KX-2000 hollow inclusion strain sensors and the KJ327-F portable digital static-resistance strain indicators were used for the field measurement [11]. The strain that the hollow inclusion sensors can measure ranges ±20 000 µε, with a precision of ±0.1%. Meanwhile, adequate compensation is taken to avoid the effect of temperature change on the measured strains.

After the probes in each monitoring drilling were installed at predetermined positions, the holes were filled with cement grout and allowed to harden for 28 days before the initial strain of the probe sensors was measured. After the actual mining officially started in the mining face, the advancement distances of the mining face and the corresponding strains of each probe sensor were recorded every day. During the field measurement, to ensure that the real-time strains were unaffected by unfavourable factors, such as coal mining and human activities, each strain datapoint of the probe sensor was measured by the maintenance crew daily at an interval of 30 min. The entire field measurement lasted for eight months.

## 4.2. *In situ* failure depth of floor rocks

By analysing the strain evolution curves of the 17 sensor probes in the four monitoring boreholes, the floor rock damage characteristics and the floor rocks failure depth are shown in table 6. It can be seen that in the mining face 100502-80, the failure depth of the floor rocks affected by mining activity is 12.50–14.65 m, and when the mining face length increases to 180 m, the failure depth is 17.50–19.20 m. The larger the mining face length, the greater the failure depth of the floor rocks affected by mining activity.

## 4.3. Model validation

To check the accuracy of this model, the *in situ* measurements in the Yitang coal mine are taken for model validation. In the no. 100 502 mining face, the mining face length is 80 m or 180 m, the coal seam pitch is 5°, the aquifer water pressure is 1.5 MPa, and the burial depth is 500 m. These *in situ* geological parameters are taken into equation (3.2).

When the mining face length is 80 m, the failure depth of floor rocks is

$$h = -45.8526 + 7.074\ln 80 + 5.504\ln 5 + 9.1214e^{0.00135\times 500} - 0.4079\ln 1.5 = 11.31\,\text{m}. \qquad (4.1)$$

When the mining face length is 180 m, the failure depth of floor rocks is

$$h = -45.8526 + 7.074\ln 180 + 5.504\ln 5 + 9.1214e^{0.00135\times 500} - 0.4079\ln 1.5 = 17.05\,\text{m}. \qquad (4.2)$$

The failure depths of model prediction and the *in situ* measurements are compared and the relative errors with mining face lengths of 80 m and 180 m are 9.52% and 2.57%, respectively, which indicates that this multifactor prediction model can accurately calculate the failure depths of floor rocks under different geological conditions.

**Table 6.** Failure and damage characteristics of the floor rocks at the burial locations of each probe sensor.

| mining face | probe sensor | buried depth (m) | failure or not | mutation type | failure depth (m) |
|---|---|---|---|---|---|
| 100502-80 | 1 | 4.93 | Y | compressive-tensile strain mutation | 12.5–14.65 |
| | 2 | 7.93 | Y | abnormal mutation | |
| | 5 | 9.5 | Y | compressive-tensile strain mutation | |
| | 3 | 10.93 | Y | abnormal mutation | |
| | 6 | 12.5 | Y | compressive-tensile strain mutation | |
| | 4 | 14.65 | N | — | |
| | 7 | 15.5 | N | — | |
| | 8 | 17.5 | N | — | |
| | 9 | 19.5 | N | — | |
| 100502-180 | 10 | 12.2 | Y | compressive-tensile strain mutation | 17.5–19.2 |
| | 15 | 13.5 | Y | compressive-tensile strain mutation | |
| | 11 | 15.2 | Y | compressive-tensile strain mutation | |
| | 12 | 17.2 | probe damaged | — | |
| | 16 | 17.5 | Y | compressive-tensile strain mutation | |
| | 13 | 19.2 | N | — | |
| | 17 | 20 | N | — | |
| | 14 | 22 | N | — | |

## 4.4. Sensitivity analysis of each impact factor in the multifactor model

Both the F-test and range analysis are conducted to analyse the sensitivity of each impact factor on the failure depth in the multifactor model.

### 4.4.1. F-test

In the F-test, the influence degree of each impact factor on the failure depth of floor rocks can be calculated by the following equation:

$$F_a = \frac{S_i/f_i}{S_e/f_e},$$

(4.3)

where $F_a$ is the F-value of each impact factor, $S_i$ is the residual sum of squares, $f_i$ is the residual degree of freedom, $S_e$ is the sum of squares of the error and $f_e$ is the degree of freedom of the error.

According to equation (4.3), the F-value of each impact factor is calculated and shown in table 7. The F-value of each impact factor is compared with the F-distribution table. The influence of coal seam pitch is of high significance for the failure depth of floor rocks, and the influence of other impact factors (mining face length, burial depth and aquifer water pressure) is significant. In addition, the influence order of the impact factors on the failure depth of floor rocks is coal seam pitch > mining face length > burial depth > aquifer water pressure.

**Table 7.** Significance analysis of the impact factors on the floor rocks. Notes: (1) $F_{0.01}(3, 9) = 6.99$, $F_{0.05}(3, 9) = 3.86$ and $F_{0.1}(3, 9) = 2.81$. (2) If $F > F_{0.01}(f_i, f_e)$, this impact factor is of high significance, and denoted by '**'. (3) If $F_{0.1}(f_i, f_e) < F < F_{0.01}(f_i - f_e)$, this impact factor is significant, and denoted by '*'. (4) If $F < F_{0.1}(f_j, f_e)$, this impact factor is not significant and denoted by '—'.

| impact factor | sum of squares | degree of freedom | mean square | $F$ | $F_a$ | significance |
|---|---|---|---|---|---|---|
| mining face length | 76.832075 | 3 | 25.61069167 | 1.91926772 | 3.86 | — |
| coal bed pitch | 131.7711 | 3 | 43.9237 | 3.291646343 | 3.86 | * |
| burial depth | 22.95912 | 3 | 7.65304 | 0.573519561 | 3.86 | — |
| aquifer water pressure | 2.646325 | 3 | 0.882108333 | 0.066105284 | 3.86 | — |
| error | 40.03197375 | 3 | 13.34399125 | | | |

**Table 8.** Range of failure depth by each impact factor.

| impact factor | range (m) |
|---|---|
| mining face length | 5.35 |
| coal bed pitch | 7.70 |
| burial depth | 3.30 |
| aquifer water pressure | 1.00 |

### 4.4.2. Range analysis

In the range analysis, a range is calculated by the difference of the maximum failure depth and the minimum failure depth affected by an individual impact factor [21]. The range analysis results of the failure depth by each impact factor are listed in table 8.

According to the range analysis results, the influence order of the impact factors on the failure depth of floor rocks is coal seam pitch > mining face length > burial depth > aquifer water pressure, which is consistent with the order results of the $F$-test. Consequently, the impact of the coal seam pitch on the failure depth of the floor rocks is the most significant, the impact of mining face length and the burial depth on the failure depth of floor rocks is less significant, and the impact of the aquifer water pressure on the failure depth of the floor rocks is the least significant.

## 5. Conclusion

(1) Orthogonal numerical tests considering mining face length, coal seam pitch, burial depth and aquifer water pressure show that the failure depth of the floor rocks increases with increasing mining face length, coal seam pitch and burial depth. A multifactor prediction model for the failure depth of floor rocks was established.

(2) In the *in situ* measurement of the Yitang coal mine, the failure depths of the floor rocks in the 100 502 (80 m) and 100 502 (180 m) mining faces were approximately 12.50–14.65 m and 17.50–19.20 m, respectively. The *in situ* measurement results were taken into the prediction model, and the model results were in good agreement with the *in situ* data.

(3) The sensitivity of each impact factor (mining face length, coal seam pitch, burial depth and aquifer water pressure) in the model on the failure depth of floor rocks was analysed by both $F$-test and range analysis, and the results show the influence order of the impact factors on the failure depth of floor rocks is coal bed pitch > mining face length > burial depth > aquifer water pressure.

Data accessibility. All data are included in the manuscript as tables and figures and have been uploaded as part of the electronic supplementary material.

Authors' contributions. Y.J. carried out the theoretical analysis and numerical simulation and wrote the manuscript. T.C. conceived and designed the framework. X.Z. supervised the work. All the authors performed the *in situ* study and approved the final manuscript.

Competing interests. The authors declare no conflict of interest.

Funding. Support for this work was provided by the National Natural Science Foundation of China (grant no. 51704204).

Acknowledgements. The authors are very grateful to the editors and reviewers for their kind and invaluable comments.

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
