## [Reviewer comments · Royal Society Open Science]

Review History

RSOS-190528.R0 (Original submission)

Review form: Reviewer 1

Is the manuscript scientifically sound in its present form?

Yes

Are the interpretations and conclusions justified by the results?

Yes

Is the language acceptable?

No

Is it clear how to access all supporting data?

Yes

Do you have any ethical concerns with this paper?

No

Have you any concerns about statistical analyses in this paper?

No

Recommendation?

Major revision is needed (please make suggestions in comments)

Comments to the Author(s)

The failure depth of the floor strata is of great significance for ensuring the safe mining over an aquifer. In this paper, a multi-factor coupling orthogonal numerical tests on failure depth of floor strata were performed, the factors impact was analyzed and a mathematics multi-factor prediction model was established. Furthermore, the prediction model was validated by the in-situ monitoring results. The results can provide important guidance to coal mining under water pressure and offer a theoretical reference for failure depth control of floor rocks under similar geophysical conditions. Thence, the paper is interesting, and the work of this manuscript is practical and logical.

There are some problems to be further improved as follows:

1. The English of the manuscript must be improved seriously. We suggest you to get assistance from colleagues who are well-versed in English.
2. There are too few overseas relevant research review in the Introduction part.
3. The words should be in unified font form in the Table 2.
4. In Fig. 2, the damage zone in the floor is not clear.
5. Does the impact factor influence order on failure depth applicable in the other similar conditions?
6. The paper should be revised following the format specification of the journal.

Review form: Reviewer 2

Is the manuscript scientifically sound in its present form?

Yes

Are the interpretations and conclusions justified by the results?

Yes

Is the language acceptable?

Yes

Is it clear how to access all supporting data?

Yes

Do you have any ethical concerns with this paper?

No

Have you any concerns about statistical analyses in this paper?

No

Recommendation?

Accept with minor revision (please list in comments)

Comments to the Author(s)

The manuscript titled "A Multi-factor Coupling Prediction Model for the Failure Depth of Floor Rocks in Fully-mechanized Caving Mining: A Numerical and In-situ Study" presents an interesting work on the failure depth prediction of floor rocks in mining through the methods of numerical tests and field monitoring. The prediction model that established in this paper works well in describing the failure depth of floor rocks in mining under different geological conditions. I would like to recommend this manuscript to be accepted for publication if the following small issues could be settled.

- 1) In the numerical tests, how did you apply the aquifer water pressure to the coal seam and why?
- 2) In Table 3, the failure depths of floor rocks in each numerical test are presented, but the data lack of proper units, which makes me confusing.
- 3) In this field monitoring, how did you tell the floor rocks fail or not?
- 4) In section 5.1, how was the strain measured? The detailed measurement principle should be clearly stated.
- 5) In the model validation, both the F-test and range test are used and what is the validation difference between the two tests?
- 6) The English editing should be improved.
- 7) The figures should be provided with high-resolution.

Decision letter (RSOS-190528.R0)

21-Jun-2019

Dear Dr Jiang,

The editors assigned to your paper ("A Multi-factor Coupling Prediction Model for the Failure Depth of Floor Rocks in Fully-mechanized Caving Mining: A Numerical and In-situ Study") have now received comments from reviewers. We would like you to revise your paper in accordance with the referee and Associate Editor suggestions which can be found below (not including confidential reports to the Editor). Please note this decision does not guarantee eventual acceptance.

Please submit a copy of your revised paper before 14-Jul-2019. Please note that the revision deadline will expire at 00.00am on this date. If we do not hear from you within this time then it will be assumed that the paper has been withdrawn. In exceptional circumstances, extensions may be possible if agreed with the Editorial Office in advance. We do not allow multiple rounds of revision so we urge you to make every effort to fully address all of the comments at this stage. If deemed necessary by the Editors, your manuscript will be sent back to one or more of the original reviewers for assessment. If the original reviewers are not available, we may invite new reviewers.

Please pay attention to the various issues raised by the reviewers. Although their comments are relatively brief, they do raise matters of substance that need to be addressed before we can consider publication. I share the reviewers and Associate Editor concerns about the use of English and suggest that you enlist a native English speaker or professional assistance to get your manuscript into good shape. Remember: if a paper is not well-written, it will not be well-read.

To revise your manuscript, log into <http://mc.manuscriptcentral.com/rsos> and enter your

Author Centre, where you will find your manuscript title listed under "Manuscripts with Decisions." Under "Actions," click on "Create a Revision." Your manuscript number has been appended to denote a revision. Revise your manuscript and upload a new version through your Author Centre.

- Data accessibility

If you wish to submit your supporting data or code to Dryad (<http://datadryad.org/>), or modify your current submission to dryad, please use the following link:
<http://datadryad.org/submit?journalID=RSOS&manu=RSOS-190528>

- Competing interests

- Authors' contributions

AB carried out the molecular lab work, participated in data analysis, carried out sequence alignments, participated in the design of the study and drafted the manuscript; CD carried out the statistical analyses; EF collected field data; GH conceived of the study, designed the study,

coordinated the study and helped draft the manuscript. All authors gave final approval for publication.

- Acknowledgements

- Funding statement

on behalf of Dr Pablo Gonzalez (Associate Editor) and Jon Blundy (Subject Editor)
openscience@royalsociety.org

Associate Editor's comments (Dr Pablo Gonzalez):

Dear Authors,

Now, we have received two revision reports back from your submitted manuscript. They seem to agree that the quality of the manuscript is good. However, at this point the manuscript still need some major revisions.

Reviewers highlighted 1) several technical aspects that need serious clarifications. And a common recommendation is a 2) strong revision of the English language. I agree with their view, and recommend that the manuscript will be sent back to you pending of major revisions. In particular, I will pay attention to the revision of the English language in any resubmission. I hope you could hire a professional or a native English speaker to improve your manuscript.

Thanks for your patience and also considering RSOS journal for your submission. I hope to see the resubmitted manuscript soon.

Best regards,
Pablo

Comments to Author:

Reviewers' Comments to Author:

Reviewer: 1

Comments to the Author(s)

The failure depth of the floor strata is of great significance for ensuring the safe mining over an aquifer. In this paper, a multi-factor coupling orthogonal numerical tests on failure depth of floor strata were performed, the factors impact was analyzed and a mathematics multi-factor

prediction model was established. Furthermore, the prediction model was validated by the in-situ monitoring results. The results can provide important guidance to coal mining under water pressure and offer a theoretical reference for failure depth control of floor rocks under similar geophysical conditions. Thence, the paper is interesting, and the work of this manuscript is practical and logical.

There are some problems to be further improved as follows:

1. The English of the manuscript must be improved seriously. We suggest you to get assistance from colleagues who are well-versed in English.
2. There are too few overseas relevant research review in the Introduction part.
3. The words should be in unified font form in the Table 2.
4. In Fig. 2, the damage zone in the floor is not clear.
5. Does the impact factor influence order on failure depth applicable in the other similar conditions?
6. The paper should be revised following the format specification of the journal.

Reviewer: 2

Comments to the Author(s)

The manuscript titled "A Multi-factor Coupling Prediction Model for the Failure Depth of Floor Rocks in Fully-mechanized Caving Mining: A Numerical and In-situ Study" presents an interesting work on the failure depth prediction of floor rocks in mining through the methods of numerical tests and field monitoring. The prediction model that established in this paper works well in describing the failure depth of floor rocks in mining under different geological conditions. I would like to recommend this manuscript to be accepted for publication if the following small issues could be settled.

- 1) In the numerical tests, how did you apply the aquifer water pressure to the coal seam and why?
- 2) In Table 3, the failure depths of floor rocks in each numerical test are presented, but the data lack of proper units, which makes me confusing.
- 3) In this field monitoring, how did you tell the floor rocks fail or not?
- 4) In section 5.1, how was the strain measured? The detailed measurement principle should be clearly stated.
- 5) In the model validation, both the F-test and range test are used and what is the validation difference between the two tests?
- 6) The English editing should be improved.
- 7) The figures should be provided with high-resolution.

Comments from Editorial Office to Authors:

For more information about language editing services endorsed by the Royal Society, please follow the link below:

<https://royalsociety.org/journals/authors/language-polishing/>

Author's Response to Decision Letter for (RSOS-190528.R0)

See Appendix A.

Decision letter (RSOS-190528.R1)

25-Jul-2019

Dear Dr Zhang:

On behalf of the Editors, I am pleased to inform you that your Manuscript RSOS-190528.R1 entitled "A Multifactor Coupling Prediction Model for the Failure Depth of Floor Rocks in Fully mechanized Caving Mining: A Numerical and In Situ Study" has been accepted for publication in Royal Society Open Science subject to minor revision in accordance with the editor's suggestions. The Associate Editor is happy for your paper to be published pending one very small correction, as noted below.

The Editors have recommended publication, but also suggest some minor revisions to your manuscript. Therefore, I invite you to respond to the comments and revise your manuscript.

- Ethics statement

- Data accessibility

If you wish to submit your supporting data or code to Dryad (<http://datadryad.org/>), or modify your current submission to dryad, please use the following link:
<http://datadryad.org/submit?journalID=RSOS&manu=RSOS-190528.R1>

- Competing interests

- Authors' contributions

- Acknowledgements

- Funding statement

Because the schedule for publication is very tight, it is a condition of publication that you submit the revised version of your manuscript before 03-Aug-2019. Please note that the revision deadline will expire at 00.00am on this date. If you do not think you will be able to meet this date please let me know immediately.

on behalf of Dr Pablo Gonzalez (Associate Editor) and Jon Blundy (Subject Editor)
openscience@royalsociety.org

Associate Editor Comments to Author (Dr Pablo Gonzalez):

Dear Authors,

Thanks for addressing the comments of the two reviewers. I will be glad to accept this publication, pending minor changes in the format of information in the section 3.2. from line 49 to 60, and section 5.4. (b) from line 45 to 50. That information will be more effectively communicated using tabular format.

Kind regards,
Pablo J. Gonzalez

Author's Response to Decision Letter for (RSOS-190528.R1)

See Appendix B.

Decision letter (RSOS-190528.R2)

30-Jul-2019

Dear Dr Zhang,

I am pleased to inform you that your manuscript entitled "A Multifactor Coupling Prediction Model for the Failure Depth of Floor Rocks in Fully mechanized Caving Mining: A Numerical and In Situ Study" is now accepted for publication in Royal Society Open Science.

on behalf of Dr Pablo Gonzalez (Associate Editor) and Jon Blundy (Subject Editor)
openscience@royalsociety.org

Follow Royal Society Publishing on Twitter: [@RSocPublishing](https://twitter.com/RSocPublishing)
Follow Royal Society Publishing on Facebook:
<https://www.facebook.com/RoyalSocietyPublishing.FanPage/>
Read Royal Society Publishing's blog: <https://blogs.royalsociety.org/publishing/>

Appendix A

Dear Dr. Gonzalez and reviewers,

Thank you for your letter and the comments concerning our manuscript entitled “**A Multi-factor Coupling Prediction Model for the Failure Depth of Floor Rocks in Fully Mechanized Caving Mining: A Numerical and In-situ Study**” (ID: RSOS-190528). All your comments are valuable and helpful for improving our paper. We have considered your recommendations and the reviewer's comments carefully and have made detailed revisions to our previous manuscript. In our revised submission, the revised parts are highlighted in red for your convenience. The responses to the Associate Editor's recommendation and the reviewer's comments are listed below.

Responses to Associate Editor:

1) **Reviewers highlighted 1) several technical aspects that need serious clarifications. And a common recommendation is a 2) strong revision of the English language. I agree with their view, and recommend that the manuscript will be sent back to you pending of major revisions. In particular, I will pay attention to the revision of the English language in any resubmission. I hope you could hire a professional or a native English speaker to improve your manuscript.**

Authors' response: Thanks for your comments. According to your suggestion, we have made detailed modifications to our previous manuscript. In our revised version, we rewrote the introduction section to highlight the innovation in our paper, a multifactor coupling prediction model for the failure depth of floor rocks in fully mechanized caving mining. Importantly, we also improved the language editing by the Springer Nature Author Service. Here is the language editing certification.

Nature Research Editing Service Certification

This is to certify that the manuscript titled **A Multi-factor Coupling Prediction Model for the Failure Depth of Floor Rocks in Fully-mechanized Caving Mining: A Numerical and In-situ Study** was edited for English language usage, grammar, spelling and punctuation by one or more native English-speaking editors at Nature Research Editing Service. The editors focused on correcting improper language and rephrasing awkward sentences, using their scientific training to point out passages that were confusing or vague. Every effort has been made to ensure that neither the research content nor the authors' intentions were altered in any way during the editing process.

Documents receiving this certification should be English-ready for publication; however, please note that the author has the ability to accept or reject our suggestions and changes. To verify the final edited version, please visit our verification page. If you have any questions or concerns over this edited document, please contact Nature Research Editing Service at support@as.springernature.com.

Manuscript title: A Multi-factor Coupling Prediction Model for the Failure Depth of Floor Rocks in Fully-mechanized Caving Mining: A Numerical and In-situ Study

Authors: Yulong Jiang, Tingting Cai, Xiaoqiang Zhang*

Key: 3C46-57C2-10FC-104E-80EE

This certificate may be verified at secure.authorservices.springernature.com/certificate/verify.

Responses to reviewer #1:

Reviewer's comments: The failure depth of the floor strata is of great significance for ensuring the safe mining over an aquifer. In this paper, a multi-factor coupling orthogonal numerical tests on failure depth of floor strata were performed, the factors impact was analyzed and a mathematics multi-factor prediction model was established. Furthermore, the prediction model was validated by the in-situ monitoring results. The results can provide important guidance to coal mining under water pressure and offer a theoretical reference for failure depth control of floor rocks under similar geophysical conditions. Thence, the paper is interesting, and the work of this manuscript is practical and logical.

1) **The English of the manuscript must be improved seriously. We suggest you to get assistance from colleagues who are well-versed in English.**

Authors' response: Thanks for your comments. We are sorry for our improper writing. The language editing in our manuscript have been improved by Springer Nature Author Service. Here is the language editing certification.

Nature Research Editing Service Certification

This is to certify that the manuscript titled *A Multi-factor Coupling Prediction Model for the Failure Depth of Floor Rocks in Fully-mechanized Caving Mining: A Numerical and In-situ Study* was edited for English language usage, grammar, spelling and punctuation by one or more native English-speaking editors at Nature Research Editing Service. The editors focused on correcting improper language and rephrasing awkward sentences, using their scientific training to point out passages that were confusing or vague. Every effort has been made to ensure that neither the research content nor the authors' intentions were altered in any way during the editing process.

Documents receiving this certification should be English-ready for publication; however, please note that the author has the ability to accept or reject our suggestions and changes. To verify the final edited version, please visit our verification page. If you have any questions or concerns over this edited document, please contact Nature Research Editing Service at support@as.springernature.com.

Manuscript title: A Multi-factor Coupling Prediction Model for the Failure Depth of Floor Rocks in Fully-mechanized Caving Mining: A Numerical and In-situ Study

Authors: Yulong Jiang, Tingting Cai, Xiaoqiang Zhang*

Key: 3C46-57C2-10FC-104E-80EE

This certificate may be verified at secure.authorservices.springernature.com/certificate/verify.

2) **There are too few overseas relevant research reviews in the Introduction part.**

Authors' response: Thanks for your comments. We cite many high-quality papers published in international journals reporting the failure of floor rocks in China because they are of great significant guidance for the failure of floor rocks control. According to your recommendation, we rewrote the introduction section and supplemented several overseas researches to enrich the literature review in our revised version. The detailed revision is listed below.

“Currently, many studies on the failure of floor rocks have been reported since a critical energy-release point was introduced to analyze the bearing capacity and stability principles of the floor rocks in long-term conditions by Santos

& Bieniawski [2,3]. Kumar and Das carried out plate loading tests on simulated floor strata for varying geotechnical conditions of weak floor strata to evaluate the bearing strength characteristics of floor strata during an excavation process while under heavy loads from an overlying rock mass [4]. The failure depth in a specific longwall gangue backfilling mine was measured using the mine electricity profiling method by Yang et al. [5]. Yin et al. performed field monitoring of the stress state of the floor rocks during the mining process and revealed the relationship between the vertical/horizontal stress increments of the floor rocks with depth [6,7]. Based on the isotropic assumption of rocks, Zhang determined the failure depth of floor rocks by comparing the flow rate of water in a borehole before and after coal mining [8]. In combination with the methods of field measurement, theoretical analysis and numerical simulation, Zhu et al. [9] and Wang et al. [10] developed analytical solutions of the support stress distribution of floor rocks to quantify the failure depths of floor rocks. Jiang et al. performed an in-situ measurement by hollow inclusion strain sensors to investigate the mining-induced damage and failure characteristics of coal seam floor rocks at different depths in working faces with different lengths [11]. Generally, among the above studies, many impact factors, such as in-situ stress, burial depth and isotropic rocks, were taken into accounts [12-15], while few studies that include aquifer water have been reported. The failure depth of the floor rocks affected by the mining process is mainly composed of two parts: the direct failure zone formed by the mining stress field and the floor rising zone affected by the aquifer water. The sum of the two is the final failure depth of the floor rocks [16]. However, the effect of aquifer water on the direct failure zone is always ignored. The high-pressure of the aquifer water will cause structural changes in the floor rocks and can eventually lead to the changes in failure depth of floor rocks [17]. Therefore, the influence of the aquifer water on the failure depth of the floor rocks cannot be ignored.

The in-situ measurement of the stress or strain of floor rocks under active mining is the most direct and effective method for determining the deformation and failure of floor rocks [18-20]. However, limited by the specific and nonrepeatable geophysical conditions in in-situ monitoring, only certain impact factors could be considered in field measurements. The failure of floor rocks is always the result of several impact factors, and numerical testing is an effective method to take multi-factor coupling into consideration. It is of great significance to study the failure depth of floor rocks affected by multiple impact factors and give the impact order of the factors.

Please see section 2 Introduction. The revisions are marked red.

3) The words should be in unified font form in the Table 2.

Authors' response: Thanks for your comments. We are sorry for our improper writing. In our revised manuscript, we have checked the entire text carefully and corrected all the similar improper writings in tables and figures.

4) In Fig. 2, the damage zone in the floor is not clear.

Authors' response: Thanks for your comments. As the review suggested, we supplied more high-resolution images as clear as possible to present our content in our revised manuscript. Please see the figures in our revised version. We

hope these figures can meet your approval.

5) Does the impact factor influence order on failure depth applicable in the other similar conditions?

Authors' response: Thanks for your comments. This is a valid issue. In our paper, the four impact factors (coal seam pitch, mining face length, burial depth and aquifer water pressure) were taken into consideration to study the failure depth of floor rocks in mining. From the 16 numerical simulation tests and in-situ field monitoring results, we came to the conclusions of the impact factor order coal seam pitch > mining face length > burial depth > aquifer water pressure. The results can provide important guidance to coal mining under water pressure and offer a key theoretical reference for failure depth control of floor rocks under similar geophysical conditions. In our revised version, we highlighted our results and meanings. Please see section 2 Introduction and section 6 Conclusions.

6) The paper should be revised following the format specification of the journal.

Authors' response: Thanks for your comment. In our revised manuscript, we have carefully read the introductions for authors and prepared our resubmission according to the format specification of the journal. We hope our corrections can meet your approval.

Responses to reviewer #2:

Reviewer's comments: The manuscript titled "A Multi-factor Coupling Prediction Model for the Failure Depth of Floor Rocks in Fully-mechanized Caving Mining: A Numerical and In-situ Study" presents an interesting work on the failure depth prediction of floor rocks in mining through the methods of numerical tests and field monitoring. The prediction model that established in this paper works well in describing the failure depth of floor rocks in mining under different geological conditions. I would like to recommend this manuscript to be accepted for publication if the following small issues could be settled.

1) In the numerical tests, how did you apply the aquifer water pressure to the coal seam and why?

Authors' response: Thanks for your comments. In our paper, the impact factors including the burial depth, the mining face length, coal seam pitch and the aquifer water pressure were taken into consideration. In the numerical tests, a three-dimensional fluid-solid coupling numerical simulation model is established. The overlying rock strata (except the coal seam roof) are uniformly loaded to the upper surface. The aquifer water pressure is loaded to the coal seam at a rate of -0.01 MPa/m downward along the coal seam. The detailed revision can be seen in section 3.1 numerical tests.

2) In Table 3, the failure depths of floor rocks in each numerical test are presented, but the data lack of proper units, which makes me confusing.

Authors' response: Thanks for your comments. In our revised version, the units of the failure depths of floor rocks in Table 3 were supplemented. Please see Table 3.

3) In this field monitoring, how did you tell the floor rocks fail or not?

Authors' response: Thanks for your comments. In the filed monitoring, the strain increment is defined as the difference between the real-time strain and the initial strain of each sensor probe, and the evolution of the strain increment is used to characterize the stress and deformation of the coal seam floor rocks before and after mining. The strain increment curves with abrupt changes could be divided into two types: tensile-compressive strain mutations and abnormal mutations. From the strain increment curves of the seventeen sensor probes in the four monitoring boreholes, the floor rocks damage characteristics are listed in Table 5. In our revised version, we supplemented the damage characteristics of floor rocks at the locations of each probe sensor to better reflect the failure depth of the floor rocks. Please see Table 5.

4) In section 5.1, how was the strain measured? The detailed measurement principle should be clearly stated.

Authors' response: Thanks for your comments. In the field monitoring, the strain sensors are buried in holes at different depths, and the strain evolutions are recorded during the mining process. When the mining face is far away from the strain sensor, and the mining pressure has not reached the sensor, the strain evolution is steady. When the sensor is affected by the mining pressure, the strain evolution changes synchronously with the advancement of mining. When the mining face is worked near the strain sensor, the strains sharply increase due to the intense disturbance of the mining pressure; specifically, if the rocks at the buried position are not yet damaged, the force of the probe is relatively uniform, and the sensor strains change regularly, fluctuating with deformation of the surrounding rocks during the entire monitoring process. Once the surrounding rocks at the buried positions are disturbed violently, strong plastic displacement and deformation of the drilling hole inevitably occur and damaged plastic coating of the probe or even malfunctions of the probe are resulted; thus, the measured strains are highly discrete or fluctuate randomly. Therefore, the field strain evolutions measured can be used to determine whether the floor rock is damaged or not. In our revised manuscript, the detailed measurement principle was supplemented and listed below,

“In field monitoring, the strain sensors are buried in holes at different depths, and the strain evolutions are recorded during the mining process. Once the surrounding rocks at the buried positions are disturbed violently, strong plastic displacement and deformation of the drilling hole inevitably occur and damaged plastic coating of the probe or even malfunctions of the probe; thus, the measured strains are highly discrete or fluctuate randomly to reflect the damaged floor rocks [11].”

Please see section 5.1 In-situ monitoring.

5) In the model validation, both the F-test and range test are used and what is the validation difference between the two tests?

Authors' response: Thanks for your comments. In our paper, both the F-test and range analysis are conducted to analyze the sensitivity of each impact factor on the failure depth in multi-factor model. We intended to show the consistent validation results of the two methods. From the model validation, we can see, the influence order of the

impact factor on failure depth of floor rocks is coal seam pitch>mining face length>burial depth>aquifer water pressure, which is consistent with the order results of F-test. There is no validation difference between the two different tests.

6) The English editing should be improved.

Authors' response: Thanks for your comment. In our revised version, the language editing in our revised manuscript has been improved by the Springer Nature Author Service. Here is the language editing certification.

Nature Research Editing Service Certification

This is to certify that the manuscript titled A Multi-factor Coupling Prediction Model for the Failure Depth of Floor Rocks in Fully-mechanized Caving Mining: A Numerical and In-situ Study was edited for English language usage, grammar, spelling and punctuation by one or more native English-speaking editors at Nature Research Editing Service. The editors focused on correcting improper language and rephrasing awkward sentences, using their scientific training to point out passages that were confusing or vague. Every effort has been made to ensure that neither the research content nor the authors' intentions were altered in any way during the editing process.

Documents receiving this certification should be English-ready for publication; however, please note that the author has the ability to accept or reject our suggestions and changes. To verify the final edited version, please visit our verification page. If you have any questions or concerns over this edited document, please contact Nature Research Editing Service at support@as.springernature.com.

Manuscript title: A Multi-factor Coupling Prediction Model for the Failure Depth of Floor Rocks in Fully-mechanized Caving Mining: A Numerical and In-situ Study

Authors: Yulong Jiang, Tingting Cai, Xiaoqiang Zhang*

Key: 3C46-57C2-10FC-104E-80EE

This certificate may be verified at secure.authorservices.springernature.com/certificate/verify.

7) The figures should be provided with high-resolution.

Authors' response: Thanks for your comments. In our revised manuscript, we supplied more high-resolution images as clear as possible to present our content. Please see the figures in our revised version. We hope these figures can meet your approval.

In the revised manuscript, many detailed contents about the numerical simulation tests and in-situ monitoring were supplied to make our manuscript clear and convincing. Thanks again for your valuable comments and advice, your deep comments and practical advice contributed a lot to improve the quality of this article.

We are looking forward to hearing from you regarding our revised manuscript. We would be glad to respond to any further questions and comments that you may have.

Best regards.

Sincerely yours,

Corresponding author: Xiaoqiang Zhang

Taiyuan University of Technology

E-mail: tyzxq2009@163.com

Appendix B

Dear Dr. Gonzalez,

Thank you for your letter and the comments concerning our manuscript entitled “**A Multi-factor Coupling Prediction Model for the Failure Depth of Floor Rocks in Fully Mechanized Caving Mining: A Numerical and In-situ Study**” (ID: **RSOS-190528**). We have considered your recommendations and made corresponding revisions to our previous manuscript. In our revised submission, the information in section 3.2, lines 49-60, and section 5.4, lines 45-50, were revised into the tabular format, as can be seen in Table 4 and Table 8. The revised parts are highlighted in red for your convenience.

Best regards.

Sincerely yours,

Corresponding author: Xiaoqiang Zhang

Taiyuan University of Technology

E-mail: tyzxq2009@163.com